# Evaluation of the use of GeneXpert MTB/RIF in a zone with high burden of tuberculosis in Thailand

**Nathakorn Pongpeeradech**[1]**, Yuthichai Kasetchareo**[2,3]**, Charoen Chuchottaworn**[4]**, Saranath Lawpoolsri**[1,5]**, Udomsak Silachamroon**[6]**, Jaranit Kaewkungwal**[1,5]*

**1** Faculty of Tropical Medicine, Department of Tropical Hygiene, Mahidol University, Bangkok, Thailand, **2** Division of AIDS Tuberculosis and Sexual transmitted diseases, Department of Health, Nonthaburi, Thailand, **3** Thonburi Health Center, Bangkok, Thailand, **4** Central Chest Institute of Thailand (CCIT), Ministry of Public Health, Nonthaburi, Thailand, **5** Faculty of Tropical Medicine, Center of Excellence for Biomedical and Public Health Informatics (BIOPHICS), Mahidol University, Bangkok, Thailand, **6** Faculty of Tropical Medicine, Department of Clinical Tropical Medicine, Mahidol University, Bangkok, Thailand

* jaranit.kae@mahidol.ac.th

**Data Availability Statement:** All relevant data are within the paper and its Supporting Information files.

## Abstract

GeneXpert MTB/RIF is a reliable molecular diagnostic tool capable of detecting *Mycobacterium tuberculosis* (MTB) and identifying genetic determinants of rifampicin (RIF) resistance. This study aimed to assess physicians' diagnostic decision-making processes for TB based on GeneXpert MTB/RIF results and how this affected the initiation of multidrug resistance (MDR) treatment. This study employed a mixed method: data were collected retrospectively from the medical records of TB patients and in-depth interviews were conducted with health-care workers in areas with a high TB burden in Thailand. A total of 2,030 complete TB records from 2 patient groups were reviewed, including 1443 suspected cases with negative smear results and 587 with high risk of MDR-TB. GeneXpert MTB/RIF was routinely used to assist the physicians in their decision-making for the diagnosis of pulmonary tuberculosis (PTB) and the initiation of MDR-TB treatment. The physicians used it as a "rule-in test" for all patients with negative chest X-rays (CXR) and smear results, to ensure timely treatment. Approximately one-fourth of the patients with negative CXR/smear and GeneXpert MTB/RIF results were diagnosed with PTB by the physicians, who based their decisions on other evidence, such as clinical symptoms, and did not use GeneXpert MTB/RIF as a "rule-out test." GeneXpert MTB/RIF proved effective in early detection within a day, thereby radically shortening the time required to initiate second-line drug treatment. Despite its high sensitivity for detecting PTB and MDR-TB, GeneXpert MTB/RIF had contradictory results (false positive and/or false negative) for 21.8% of cases among patients with negative smear results and 41.1% of cases among patients with high risk of MDR-TB. Therefore, physicians still used the results of other conventional tests in their decision-making process. It is recommended that GeneXpert MTB/RIF should be established at all points of care and be used as the initial test for PTB and MDR-TB diagnosis.

**Funding:** The authors received no specific funding for this work.

**Competing interests:** The authors have declared that no competing interests exist.

**Abbreviations:** AFB, Acid- Fast Bacilli; CDC, Centers for Disease Control and Prevention; CXR, chest X-ray; DST, drug susceptibility testing; HbA1, glycosylated hemoglobin; HIV, human immunodeficiency virus; IQR, interquartile range; LPA, line probe assay; MDR, multidrug resistance; MDR-TB, multidrug resistance tuberculosis; MTB, Mycobacterium tuberculosis; NAAT, nucleic acid amplification test; PTB, pulmonary tuberculosis; RIF, Rifampicin; RR, Rifampicin resistance; TAD, treatment after default; TAT, turnaround time; TrT, treatment time; WHO, World Health Organization; XDR-TB, extensively drug-resistant tuberculosis.

## Introduction

The World Health Organization (WHO) estimated that a third of the world's population has been infected with *Mycobacterium tuberculosis* (MTB) even after the development of high-tech screening tools and advances in the treatment of tuberculosis (TB), especially in the developing world [1–3]. The prevalence of TB has also been rising in a group of presumptive cases with negative smears in countries with high HIV burden, where TB mutations causing drug-resistant TB have been increasing [4,5]. Delayed diagnosis is a significant cause of increased mortality in TB patients, especially HIV patients with smear-negative TB tests [5–7]. Prompt and adequate TB diagnosis is essential for optimal TB control strategies, resulting in the early treatment of patients with TB and multidrug-resistant tuberculosis (MDR-TB).

The clinical features of TB are nonspecific, so that rapid, simple, and accurate diagnostic tools for patients at high risk of TB and MDR-TB, and those with smear-negative pulmonary tuberculosis (PTB), have been recommended [2,8]. Molecular tests for TB and MDR-TB have developed remarkably owing to the major challenges faced by countries with high TB burden, the emergence of MDR-TB, and highly drug-resistant tuberculosis (XDR-TB) worldwide [8–11]. Although the nucleic acid amplification test (NAAT) has been recommended for the routine evaluation of TB in patients since 1996, this recommendation has not been implemented widely. Since December 2010, the WHO has endorsed a novel molecular test, GeneXpert MTB/RIF, be used for the diagnosis of TB and TB resistance in TB-endemic countries. GeneXpert MTB/RIF is an automated diagnostic test in which cartridge-based NAAT is used for rapid TB diagnosis in addition to a rapid antibiotic sensitivity test. MTB DNA and rifampicin (RIF) resistance (RR) can be identified concurrently with this test [12,13] These new technologies allow better and early diagnosis of TB and RR as this tool can identify the gene mutation responsible for RR [11,14,15]. This can lead to the timely initiation of an appropriate anti-TB regimen [10,12,16].

The use of GeneXpert MTB/RIF was implemented in Thailand in 2013. However, the link between newly diagnosed patients and rapid treatment initiation has not been clearly defined since implementation. It was observed that the treatment is still largely defined by the clinician's decision. Information regarding the diagnostic accuracy of GeneXpert might be insufficient to drive its adoption by physicians in their decision-making. There have been very few reports on the impact of this novel test on TB patient outcomes in Thailand [17,18]. Therefore, this study aimed to assess the impact of GeneXpert MTB/RIF in Thailand, focusing on physicians' decision-making processes in the diagnosis of TB, particularly among patients with negative smear results, and how this affects the initiation of MDR treatment among those at high risk of MDR-TB. Specifically, this study aimed to assess the impact of GeneXpert MTB/RIF in terms of the following: (1) the diagnosis of PTB among patients with negative smear results and of TB resistance among those at high risk of MDR-TB, (2) the performance of GeneXpert MTB/RIF in detecting TB and drug-resistant TB compared with conventional methods, and (3) the turnaround time (TAT) and treatment time (TrT) using GeneXpert MTB/RIF compared with those using conventional methods, such as chest X-ray (CXR), drug susceptibility testing (DST), and line probe assay (LPA).

## Materials and methods

### Study design and case definition

The study employed a mixed method: Part I was a retrospective cohort study, and Part II a qualitative study. For Part I, data were collected retrospectively from all patients for whom GeneXpert MTB/RIF was performed among cases of suspected TB with negative sputum-

smear results and at high risk of MDR-TB. For Part II, data were collected prospectively by in-depth interviews with physicians and nurses, who worked in TB clinics in the same study areas included in Part I.

In this study, "suspected TB case" was defined according to the guidelines of the National TB Control Programme—Thailand [19]. According to the guidelines, suspected TB cases were considered from several conditions, regardless of the presence of signs and symptoms of TB. The high risk of TB involves those with close contact with pulmonary TB smear-positive or family members of active pulmonary TB, HIV patients, prisoners, diabetes patients with glycosylated hemoglobin (HbA1C) results > 5, the elderly (>65 years) with underlying diseases, who would be screened by CXR. Among this group, patients with abnormal CXR were followed up by acid-fast bacillus (AFB) sputum test and GeneXpert. In patients who were hospitalized by other diseases, but whose CXR appeared abnormal, were also screened for PTB.

## Study areas

This study was conducted at the TB clinics of four general hospitals in areas with high TB burden in northeastern Thailand. The four provinces were chosen to represent areas with high numbers of TB notification cases, even though the cure rates were high. The four study areas reported high numbers of cases, ranging between 96–110+ cases/ 100,000 during 2009–2014 [20]. GeneXpert MTB/RIF has been established for TB detection at the point of care in the selected hospitals since September 2015.

## Study population and sample sizes

For Part I, the study population included patients who were screened using GeneXpert MTB/RIF in the study areas since its establishment in September 2015. The target population included the cases of suspected PTB who had negative smear results and all patients with resistant TB, which comprised the high-risk groups for MDR-TB. Patients without complete results of the microbiological test performed on the same day [both GeneXpert MTB/RIF and AFB staining], patients aged <15 years, and patients undergoing TB treatment at the time of GeneXpert MTB/RIF testing were excluded.

Sample sizes for Part I were calculated according to the primary objectives of the study. The proportion of case notifications as PTB with a negative new smear in the study areas was 28.0%, and the incidence of high risk MDR-TB was 56.2%. With a confidence interval of 95% and a margin of error of 5%, the minimum required sample sizes for the negative new smear and the high risk of MDR-TB groups were 310 and 378, respectively. The total sample size for this primary objective was at least 700 cases (310 + 378). In order to assess the performance of GeneXpert MTB/RIF in detecting PTB and MDR-TB among smear-negative patients, the sample sizes were calculated using the formula for sensitivity and specificity of the test. Based on literature review, the sensitivity and specificity of GeneXpert MTB/RIF for detecting MDR-TB among smear-negative specimens were reported as 72.5% and 99.2%, respectively [21], whereas the estimated MDR-TB cases in Thailand that were detected and notified were 12% [22]. Thus the minimum sample size required for this objective was 1915. Overall, for Part I, 2000 patients' records were reviewed, and proportionality was applied. Based on the data from TB systems, two study sites had TB notification rates of 96–109/100000, whereas the other two sites had TB notification rates of 110+/100000; thus, data were extracted from the four study sites with the proportion of case notifications of 1:1:1.15:1.15 (465, 465, 535, and 535 records).

For Part II, the target population comprised healthcare workers (i.e., physicians and nurses) who worked in TB clinics. All physician and nurses who worked in TB clinics in the study areas were included. A total of 11 physicians and 10 nurses participated in the study.

## Data collection procedures

Data for Part 1 were extracted retrospectively from paper and digital data sources during September 2015 to August 2018 in the four study hospitals. All records with TB and MDR-TB screening by the GeneXpert MTB/RIF were reviewed and extracted from different data collection forms used in Thailand in paper-based and electronic databases. The data fields for the data extraction form were designed to answer whether the GeneXpert MTB/RIF or conventional laboratory methods would influence the physician's decision when diagnosing TB among smear-negative pulmonary TB and MDR-TB among those at high risk of MDR-TB. The data fields included those related to the final physician's diagnosis, laboratory results (of both GeneXpert MTB/RIF and conventional laboratory methods), and time gaps (Time 1-from the GeneXpert MTB/RIF performed until the result was obtained, Time 2 –from result to initiation of MDR-TB treatment, Time 3 –from time sputum sent for TB culture/sensitivity test and DST, or the other methods, to result obtained). Other extracted data included changes in anti MDR-TB drugs. The ethics committees of the Faculty of Tropical Medicine, Mahidol University and the four study sites waived the requirement for informed consent for Part I of the study, given that the data were extracted by authorized persons at the study sites. The data from data extraction were fully anonymized and aggregated before the researchers accessed them.

Data for Part II were based on in-depth interviews with healthcare workers (physicians, nurses, and public health scholars). The interview guidelines comprised open-ended questions related mainly to 3 themes: the advantages, disadvantages, and limitations, of using GeneXpert MTB/RIF for MDR-TB diagnosis. Data were recorded using an audio recorder and note-taking. The interviews normally took around 20 minutes. Content analysis was based on the main theme of the extracted information. The first author was the interviewer for all participants. Written informed consent was obtained from all participants prior to the interview.

## Data analysis

The use of GeneXpert MTB/RIF or conventional methods in the physicians' decision-making process regarding the diagnosis of TB in the smear-negative group were compared. Similarly, TB resistance among those with high risk of MDR-TB was compared. Quantitative variables were expressed as mean ± standard deviation or median, and comparisons were performed using Student's t-test and/or Mann–Whitney U-test, according to the distribution of the data. Qualitative variables were expressed as percentages and compared using the $\chi^2$ and/or Fisher's exact tests. The qualitative data from the in-depth interview was summarized using qualitative content analysis. In the contents analysis, the participant's responses to the open-ended questions were interpreted to support the Part I results and the hypotheses of this study.

## Ethical considerations

The study protocol was reviewed and approved by the Ethical Review Committee of the Faculty of Tropical Medicine, Mahidol University, Thailand, Defense Surin Hospital Ethics Committee, Yasothorn Hospital Ethics Committee, Roi-et Hospital Ethics Committee, and Sisaket Hospital Ethics Committee.

# Results

## General information of the study population

A total of 2072 records of suspected cases from both groups of participants, i.e., the suspected cases with smear-negative sputum and those with a high risk of MDR-TB, were reviewed. All

patients had records of the physicians' decision-making on TB and/or TB resistance diagnosis based on the GeneXpert MTB/RIF test and conventional clinical and laboratory tests (CXR and sputum direct smear). Of the 2027 records of suspected cases, 42 (2.1%) had indeterminate results and were excluded from the study. A total of 2030 suspected cases were included into this study. Among the study patients, 1403 (69.1%) were male and 627 (30.9%) female. Their ages ranged from 15 to 92 years, with a mean age of 52.39 ± 13.96 years. The majority of the suspected cases (532 cases, 26.21%) were within the age range 55–64 years. Almost all of the study patients lived in the four provinces of the study sites (50.2% urban areas and 48.6% rural areas) and about 1.2% were patients from other neighboring provinces. The 2030 suspected cases were divided into two subgroups: 1443 suspected cases with negative smears and 587 with high risk of MDR-TB.

## Utilization of GeneXpert MTB/RIF for PTB diagnosis among suspected cases with negative smears

There were 1443 suspected cases with negative smears. Among these, 963 cases (66.7%) were males aged 15–92 years, with a mean age of 52.28 ± 13.98 years. The majority of the patients in this group were aged 41–60 years. Regarding the final diagnosis, 740 (51.3%) were diagnosed as non-TB, 259 (18.0%) as new PTB with negative bacterial confirmation (B-), 15 (1.0%) as MDR-TB or TB resistance, and 393 (27.2%) as new PTB with positive bacterial confirmation (B+). Further, 22 (1.5%) cases were diagnosed as relapse, 12 (0.8%) as treatment after default (TAD), and 2 (0.1%) as treatment failure. About half of the patients, i.e., 776 (53.8%) patients, had no underlying diseases, whereas 333 (23.1%) had diabetes and other underlying conditions, such as chronic renal failure and hypertension, or were previously infected TB cases. Among these suspected cases with negative smears, 7.8% of the patients had normal CXR, 74.3% had CXR showing infiltration, 10.5% had CXR showing a pulmonary cavity, and 7.4% had CXR showing other abnormalities. Among these suspected cases, 78.2% had signs and symptoms related to TB, whereas 21.8% had no signs or symptoms related to TB.

Fig 1 summarizes the physicians' decision-making process for the diagnosis of TB/TB resistance among suspected cases with negative smears, in association with the use of GeneXpert MTB/RIF. Among the 1443 suspected cases with negative smears, all had undergone CXR screening. As shown in Fig 1, 113 (7.8%) cases had normal CXR (box a), 106 (7.3%) had abnormal CXR not likely to be TB (box b), 152 (10.5%) had cavity (box c), and 1072 (74.4%) had lung infiltration (box d). The patients in all boxes had been tested with direct smear for at least two specimens and all had negative results for both specimens. Based on GeneXpert MTB/RIF results, MTB was detected in 449 (31.1%) and not detected in 994 (68.9%) patients. Among the 449 patients in whom MTB was detected, 448 (99.8%) were treated by the physicians as PTB cases, and among the 994 patients in whom MTB was not detected, 255 (25.7%) were treated as PTB cases. Fig 1 presents the details of each subgroup for which the physicians still treated the cases as active TB, by using not only GeneXpert MTB/RIF results but also other evidence, such as CXR and clinical symptoms.

Table 1 shows the comparisons between the diagnosis results of GeneXpert and CXR among suspected cases with negative smears. CXR results were differentiated into four groups: normal, infiltration, cavity, and other abnormalities. The results from GeneXpert MTB/RIF were classified into three groups based on the status of MTB and RR: infection but no resistance (MTB+RR-), no infection (MTB-), and infection and resistance (MTB+RR+). Cross-tabulation between the two types of diagnosis results revealed significant differences ($P < 0.001$).

Among the 113 cases with normal CXR, 101 cases were diagnosed as non-TB (MTB-) and 12 as PTB (8 MTB+, 4 MTB-) via GeneXpert MTB/RIF.

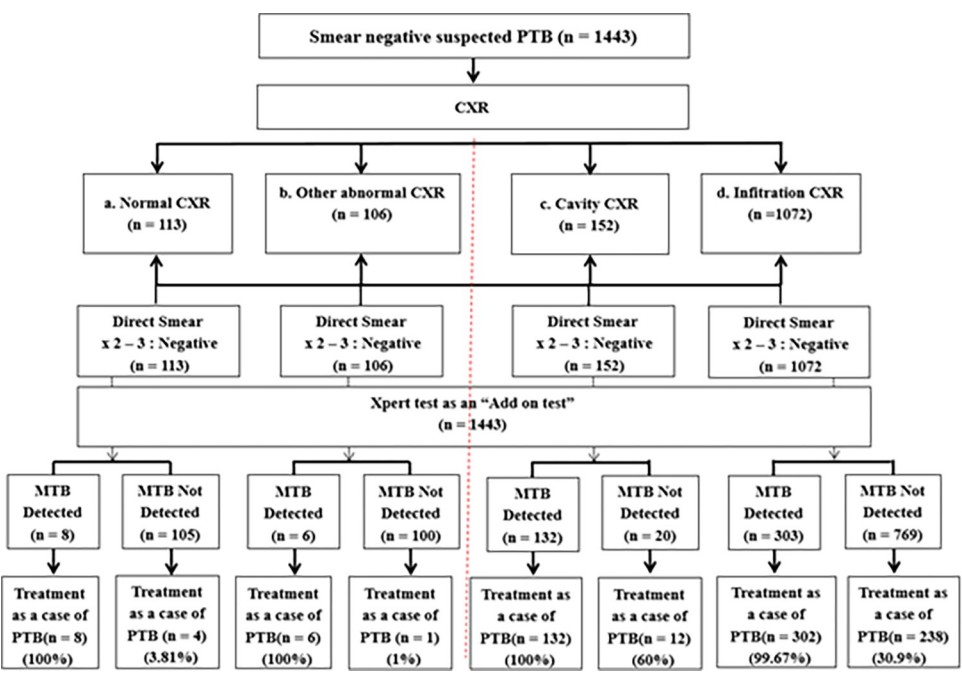

**Fig 1. Decision for types of treatments among smear-negative TB suspected cases.**

Among the 1072 cases with CXR showing infiltration, 532 cases were diagnosed as non-TB, but only 531 among these were detected via GeneXpert MTB/RIF as non-TB (MTB-). Further, 13 cases were diagnosed as MDR-TB via CXR and (MTB+RR+) GeneXpert MTB/RIF. Among

**Table 1. Comparisons of TB/MDR-TB diagnosis outcomes between CXR and GeneXpert MTB/RIF among smear negative (n = 1,443).**

| TB Diagnosis with CXR types | GeneXpert MTB/RIF (n = 1,443) | | | | p-value |
|---|---|---|---|---|---|
| | MTB⁻ | MTB +RR - | MTB +RR + | Total | |
| (Normal CXR, n = 113) | | | | | |
| Non TB | 101 (89.3%) | 0 (0.0%) | - | 101 (89.3%) | .000 |
| Pulmonary TB | 4 (3.5%) | 8 (7.1%) | - | 12 (10.6%) | |
| Total | 105 (92.9%) | 8 (7.1%) | - | 113 (100%) | |
| (Infiltration CXR, n = 1,072) | | | | | |
| Non TB | 531 (49.5%) | 1 (0.0001%) | - | 532 (49.6%) | .000 |
| MDR-TB | 0 (0.0%) | 0 (0.0%) | 13 (1.2%) | 13 (1.2%) | |
| Pulmonary TB | 238 (22.2%) | 286 (26.7%) | 3 (0.002%) | 527 (49.2%) | |
| Total | 769 (71.7%) | 287 (26.8%) | 16 (1.5%) | 1,072 (100.0%) | |
| (Cavity CXR, n = 152) | | | | | |
| Non TB | 8 (5.3%) | 0 (0.0%) | 0 (0.0%) | 8 (5.3%) | .000 |
| MDR-TB | 0 (0.0%) | 0 (0.0%) | 1 (0.01%) | 1 (0.01%) | |
| Pulmonary TB | 12 (7.9%) | 131 (86.2%) | 0 (0.0%) | 143 (94.1%)) | |
| Total | 20 (13.2%) | 131 (86.2%) | 1 (0.7%) | 152 (100.0%) | |
| (Other abnormal CXR, n = 106) | | | | | |
| Non TB | 99 (93.4%) | 0 (0.0%) | 0 (0.0%) | 99 (93.4%) | .000 |
| MDR-TB | 0 (0.0%) | 0 (0.0%) | 1 (0.9%) | 1 (0.9%) | |
| Pulmonary TB | 1 (0.9%) | 5 (4.7%) | 0 (0.0%) | 6 (5.6%) | |
| Total | 100 (94.3%) | 5 (4.7%) | 1 (0.9%) | 106 (100.0%) | |

the 527 cases of PTB diagnosed via CXR, 286 were classified via GeneXpert MTB/RIF as MTB +RR-, 238 as MTB-, and 3 as MTB+RR+.

Among the 152 cases with CXR showing cavity, all 8 non-TB cases were diagnosed via GeneXpert MTB/RIF as MTB-, and 1 MDR-TB case was diagnosed via GeneXpert MTB/RIF as MTB+RR+. Of the 143 cases diagnosed with PTB, 131 were classified via GeneXpert MTB/RIF as MTB+RR- and 12 as MTB-.

Among the 106 cases with CXR showing other abnormalities, all 99 non-TB cases were diagnosed as non-TB (MTB-) via GeneXpert MTB/RIF, whereas 1 MDR-TB case was diagnosed as MTB+RR+, and 6 PTB cases showed different results (5 were MTB+RR- and 1 was MTB-).

## Utilization of GeneXpert MTB/RIF for MDR-TB diagnosis among those at high risk of MDR-TB

Among 587 cases classified in the high-risk group for MDR-TB, 71.7% were males with an average age of 52.1 ± 13.9 years. When differentiating the high risk types for MDR-TB, 29.0% were of MDR-TB household contact, 22.2% TB relapse, 17.2% prisoners, 12.1% HIV coinfection, 11.4% TAD, 6.5% treatment failure, and 1.7% sputum reversion. There were 64 cases diagnosed with MDR-TB, among which 54 were cases of first diagnosis. The majority of this group had no underlying diseases (31.1%); the reported underlying diseases included previously infected with TB (25.6%), diabetes (24.4%), hypertension (2.2%), and chronic renal failure (4.6%). About 87.4% had signs and symptoms related to TB.

Of the 587 cases in the high-risk group for MDR-TB, 20 (3.4%) had normal CXR, 465 (79.2%) had abnormal CXR showing infiltrations, 76 (12.9%) had CXR showing cavity, and 26 (4.4%) had CXR showing other abnormalities. Regarding the AFB test results, 408 (69.5%) had positive AFB test results, whereas 179 (30.5%) had negative AFB test results. As shown in Fig 2, the 587 high-risk cases were divided into two groups: patients with positive initial sputum smears (n = 408) and patients with negative initial sputum smears (n = 179). GeneXpert MTB/

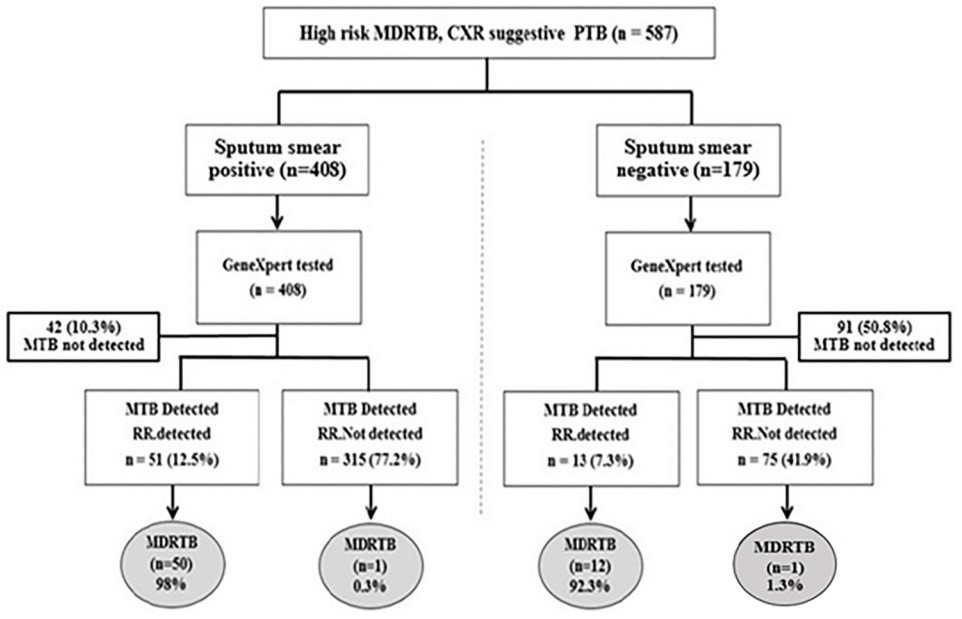

**Fig 2. Decisions in TB diagnosis among high-risk MDR-TB participants.**

RIF was used for screening in both these groups. All cases were subsequently tested with either LPA or conventional methods (culture and DST).

Of the 408 smear-positive cases as detected via GeneXpert MTB/RIF, 51 (12.5%) were MTB+RR+, 315 (77.2%) were MTB+RR-, and 42 (10.3%) were MTB not detected. Based on subsequent tests, 50 out of 51 MTB+RR+ cases (98.0%) and 1 out of 315 MTB+RR- cases (0.3%) were confirmed to be MDR-TB. According to GeneXpert MTB/RIF results, among 179 smear-negative cases, 13 (7.3%) were MTB+RR+, 75 (41.9%) were MTB+RR-, and 91 (50.8%) were MTB not detected. Based on subsequent tests, 12 (92.3%) of 13 MTB+RR+ cases and 1 (1.3%) of 75 MTB+RR- cases were confirmed as MDR-TB.

Table 2 shows the comparisons between the diagnosis results of GeneXpert MTB/RIF and CXR among cases in the high-risk group for MDR-TB. Cross-tabulation between the two types of diagnosis results revealed significant differences ($P < 0.001$).

Among 39 patients with normal CXR, out of 19 non-TB cases, 16 were detected via GeneXpert MTB/RIF as MTB- and 3 as MTB+RR-. For those with PTB, 5 and 15 out of 20 cases were detected as MTB- and MTB+RR-, respectively.

Among patients with CXR showing infiltration, 20 of 22 non-TB cases were diagnosed as MTB- and 2 as MTB+RR- via GeneXpert MTB/RIF. In this group, among 355 cases diagnosed as PTB, 289 cases were MTB+RR-, 65 were MTB-, and 1 was MTB+RR+ based on GeneXpert MTB/RIF. There were 45 cases of MDR-TB that had CXR showing infiltration: 43 were detected as MTB+RR+ and 2 as MTB+RR- via GeneXpert MTB/RIF. There were 8 patients in this group for whom the physicians changed the diagnosis to MDR-TB according to MTB+RR+ detected via GeneXpert MTB/RIF.

Among the 78 patients with CXR showing cavity, all 4 non-TB cases were detected via GeneXpert MTB/RIF as MTB-. There were 64 PTB cases, of which 51 were MTB+RR-, 12 were

**Table 2. Comparisons of TB diagnosed among high risk of MDR-TB based on CXR and GeneXpert results (n = 587).**

| TB Diagnosis with CXR types | GeneXpert MTB/RIF (n = 587) | | | | p-value |
|---|---|---|---|---|---|
| | MTB⁻ | MTB+RR- | MTB+RR+ | Total | |
| (Normal CXR, n = 39) | | | | | |
| Non TB | 16 (41.0%) | 3 (7.7%) | | 19 (48.7%) | .000 |
| Pulmonary TB | 5 (12.8%) | 15 (38.5%) | | 20 (51.2%) | |
| Total | 21 (53.8%) | 18 (46.2%) | | 39 (100.0%) | |
| (Infiltration CXR, n = 430) | | | | | |
| Non TB | 20 (4.7%) | 2 (0.5%) | 0 (0.0%) | 22 (5.1%) | .000 |
| MDRTB | 0 (0.0%) | 2 (0.5%) | 43 (10.0%) | 45 (10.5%) | |
| Pulmonary TB | 65 (15.1%) | 289 (67.2%) | 1 (0.2%) | 355 (82.5%) | |
| Change Dx (MDRTB) | 0 (0.0%) | 0 (0.0%) | 8 (1.8%) | 8 (1.8%) | |
| Total | 85 (19.8%) | 293 (68.1%) | 52 (12.1%) | 430 (100.0%) | |
| (Cavity CXR, n = 78) | | | | | |
| Non TB | 4 (5.1%) | 0 (0.0%) | 0 (0.0%) | 4 (5.1%) | .000 |
| MDRTB | 0 (0.0%) | 0 (0.0%) | 8 (10.3%) | 8 (10.3%) | |
| Pulmonary TB | 12 (15.4%) | 51 (65.4%) | 1 (1.3%) | 64 (82.1%) | |
| Change Dx (MDRTB) | 0 (0.0%) | 0 (0.0%) | 2 (2.6%) | 2 (2.6%) | |
| Total | 16 (20.5%) | 51 (65.4%) | 11 (14.1%) | 78 (100%) | |
| (Other abnormal CXR, n = 40) | | | | | |
| Non TB | 0 (0.0%) | 1 (2.5%) | 0 (0.0%) | 1 (2.5%) | .017 |
| MDRTB | 0 (0.0%) | 0 (0.0%) | 1 (2.5%) | 1 (2.5%) | |
| Pulmonary TB | 11 (27.5%) | 27 (67.5%) | 0 (0.0%) | 38 (95%) | |
| Total | 11 (27.5%) | 28 (70%) | 1 (2.5%) | 40 (100%) | |

MTB-, and 1 was MTB+RR+ based on GeneXpert MTB/RIF. Within this group, 8 MDR-TB cases were detected via GeneXpert MTB/RIF as MTB+RR+. In 2 cases, the physician changed the diagnosis to MDR-TB according to the GeneXpert MTB/RIF results of MTB+RR+.

Among 40 patients with CXR showing other abnormalities, 1 non-TB case was detected via GeneXpert MTB/RIF as MTB+RR-. Within this group, out of 38 PTB cases, 27 and 11 were detected as MTB+RR- and MTB-, respectively.

**Performance of GeneXpert MTB/RIF compared with conventional methods.** The results of GeneXpert MTB/RIF which is molecular laboratory test results compared to the phenotypic conventional method (TB culture, DST and LPA) were shown in Table 3. Among the smear-negative for TB diagnosis group that were tested with TB culture and LPA (n = 536), the GeneXpert MTB/RIF results showed discordant for 117 (21.8%) cases, and the diagnosis accuracy was 419/536 (78.2%). Given the conventional method results as gold standard, the GeneXpert MTB/RIF showed 85.1% (291/342) sensitivity, 128/194 (66.0%) specificity, 291/357 (81.5%) positive predictor value, and 128/179 (71.5%) negative predictive value (Table 3(A)).

Regarding the MDRTB diagnosis that was tested with DST and LPA (n = 355), the GeneXpert MTB/RIF results showed discordant for 41.1%) (146 cases), and the diagnosis accuracy was 209/355 (58.9%). Given the conventional method results as gold standard, the GeneXpert MTB/RIF showed 46/46 (100.0%) sensitivity, 163/309 (52.8%) specificity, 46/192 (24.0%) positive predictor value, and 163/163 (100.0%) negative predictive value (Table 3(B)).

## TAT and TrT for smear-negative and high-risk groups

The TAT for physicians' diagnosis and the TrT of TB and MDR-TB treatments were calculated for molecular and conventional methods. TAT was the timespan from the date of sputum submission to the laboratory until the date of the reported result, whereas TrT was the time elapsed between reporting of laboratory results to the TB clinic to the time of initiation of MDR-TB treatment. Not all the specimens that were detected via GeneXpert MTB/RIF were referred to be tested by conventional methods.

Table 4(A) shows TAT and TrT of GeneXpert MTB/RIF (n = 1443), TB culture (n = 522), and DST (n = 127) in the smear-negative group. The median TAT of GeneXpert MTB/RIF was 20 hours or 1 day (IQR 8–24 hours), that of conventional culture was 1296 hours or 54 days (IQR 1080–1512 hours or 45–63 days), and that of conventional DST was 1464 hours or 61 days (IQR 1386–1872 hours or 58–78 days). The median TrT of GeneXpert MTB/RIF was 4 hours or 1 day (IQR 3–6 hours), that of conventional culture was 1104 hours or 46 days (IQR

**Table 3. Diagnostic performance of GeneXpert MTB/RIF.**

**(a) TB/MDR-TB diagnosis among in smear-negative (N = 536)**

| GeneXpert MTB/RIF results | TB Culture & LPA results | | Total |
|---|---|---|---|
| | Disease + | Disease - | |
| Disease + | 291 (54.3%) | 66 (12.3%) | 357 |
| Disease - | 51 (9.5%) | 128 (23.9%) | 179 |
| Total | 342 | 194 | 536 |

**(b) MDRTB diagnosis (N = 355)**

| GeneXpert MTB/RIF results | DST & LPA results | | Total |
|---|---|---|---|
| | MDR-TB | Non MDR-TB | |
| RR detected | 46 (13.0%) | 146 (41.1%) | 192 |
| RR non detected | 0 (0.0%) | 163 (45.9%) | 163 |
| Total | 46 | 309 | 355 |

**Table 4. Turnaround time (TAT) and Treatment time (TrT) for laboratory test results and treatment initiation.**

| TAT and TrT (Hours) | Median (IQR) |
|---|---|
| (a) Smear-negative group | |
| GeneXpert TAT | 20 (8–24) |
| TB culture TAT | 1296 (1080–1512) |
| DST TAT | 1464 (1386–1872) |
| GeneXpert TrT | 4 (3–6) |
| TB culture TrT | 1104 (582–1506) |
| DST TrT | 978 (516–1718) |
| (b) High risk cases of MDR-TB | |
| GeneXpert TAT | 7 (5–16) |
| TB culture TAT | 1080 (774–1440) |
| DST TAT | 1204 (648–1680) |
| GeneXpert TrT | 6 (5–7) |
| TB culture TrT | 1488 (534–1800) |
| DST TrT | 480 (510–1723) |

582–1506 hours or 24–63 days), and that of conventional DST was 978 hours or 41 days (IQR 516–1718 hours or 22–72 days).

Table 4(B) shows TAT of GeneXpert MTB/RIF (n = 587), TB culture (n = 418), and DST (n = 286) in the high-risk group of MDR-TB. The median TAT of GeneXpert MTB/RIF was 20 hours or 1 day (IQR 8–24 hours), that of conventional culture was 1296 hours or 54 days (IQR 1080–1512 hours or 45–63 days), and that of conventional DST was 1464 hours or 61 days (IQR 1386–1872 hours or 58–78 days). The median TrT of GeneXpert MTB/RIF was 6 hours or 1 day (IQR 5–7 hours), that of conventional culture was 1488 hours or 62 days (IQR 534–1800 hours or 22–75 days), and that of conventional DST was 480 hours or 20 days (IQR 510–1723 hours or 21–72 days).

## Opinions regarding the use of GeneXpert MTB/RIF for PTB and MDR-TB diagnosis

Content analysis was performed on the information gathered from in-depth interviews with physicians and nurses who worked in TB clinics in the hospitals from the four study areas. The themes of the in-depth interviews were predefined to gain insights regarding physicians' and nurses' opinions on the efficiency of GeneXpert MTB/RIF and its utilization for diagnosis in terms of its advantages, disadvantages, and limitations.

**Advantages.** All physicians agreed that GeneXpert MTB/RIF was an efficient tool. Most of them agreed that it was beneficial for the diagnosis of MDR-TB and RR and for managing patients at high risk of MDR-TB. All healthcare workers interviewed also mentioned that the novel machine has high potential to increase TB diagnosis among smear-negative patients. The physicians and nurses also expressed their satisfaction about the faster initiation of treatment.

"... *The GeneXpert is helpful for MDR-TB prediction and prompt start of initial MDR-TB treatment.*"

"...*The GeneXpert could help as a tool to reveal the TB strains even when the direct smear is negative, so we do not miss a case.*"

". . .It is such a great facilitation tool to be used to confirm TB diagnosis among CXR abnormal cases."

"The results of GeneXpert make the process of diagnosis smooth because it takes only a few hours to get the results, so patients do not need to spend a long time in the hospital during the process of diagnosis."

"I prefer to use GeneXpert assay for TB resistant guidance treatment if it has shown rifampicin resistance rather than waiting for the DST results."

**Disadvantages.** Although all physicians agreed about the advantages of using GeneXpert MTB/RIF, half of them expressed similar thoughts that the GeneXpert MTB/RIF assay cannot be used without confirmation by conventional laboratory tests. There were cases in which the physicians found that GeneXpert MTB/RIF provided conflicting results among different test results. A few of them mentioned that they would prefer to make decisions based on the results of both GeneXpert MTB/RIF and conventional tests. A few physicians strongly commented on the view that GeneXpert MTB/RIF could not be used without confirmation from conventional tests when monitoring TB treatment.

"We cannot use only the GeneXpert alone for certain MDR-TB diagnosis as it is used only in the beginning for initial TB resistance treatment so that it could reduce the spread of the mutated strain of TB."

"I found false-positive results of rifampicin resistance from the GeneXpert test a few times."

". . .the Xpert assay can detect even the dead fragments of TB."

**Limitations.** All of the healthcare workers interviewed had similar opinions regarding the modules of GeneXpert MTB/RIF needed. Most physicians and nurses thought that there should be GeneXpert MTB/RIF with 16 modules in place at the point of care due to the large number of specimens sent from general and community hospitals to their testing facilities. Some healthcare workers noted equipment failures and long waiting times for the replacement of GeneXpert MTB/RIF while working at the local point-of-care clinic. Some physicians and nurses commented on the lack of laboratory technical support and high specimen loads from community hospitals, which in turn led to delays in obtaining test results.

". . .more machines are needed for testing specimens. . .the GeneXpert machines should be established at all points of cares with 16 modules."

". . .most hospitals have GeneXpert MTB/RIF machines with only four modules with a maximum capacity of only 12 tests per day. . .. This would affect the specimen quantity and quality. . . there had been situations when many specimens were awaiting testing and thus led to longer period time of the GeneXpert MTB/RIF results to come out."

"Delayed TATs did occur when we experienced an overload of sputum specimens sent from other nearby hospitals to us for diagnosis support and we had insufficient laboratory technicians to work on those specimens."

## Discussion

This study employed a mixed method approach by reviewing medical records and qualitative data from interviews to assess the utilization of GeneXpert MTB/RIF among healthcare

workers in Thailand. The main focus was placed on the physicians' decision-making process regarding the diagnosis of TB/MDR-TB among smear-negative patients and on the initiation of MDR treatment among those at high risk of MDR-TB. As the performance and TAT of GeneXpert MTB/RIF and conventional tests might influence the physicians' decision-making process, the study explored the sensitivity and specificity, as well as the TAT and TrT, of GeneXpert MTB/RIF and conventional tests.

Prior to the implementation of GeneXpert MTB/RIF, the WHO-recommended guidelines for TB diagnosis among smear-negative patients were follows: (1) patients with signs and symptoms of TB; (2) suspected cases with abnormal CXR; (3) patients with at least two negative sputum smears; and (4) antibiotic treatment was attempted (as a case of non-TB) for at least 2 weeks, but resulted in no improvement [23,24]. Since its introduction in 2012, GeneXpert MTB/RIF has been recommended due to its fast and simple performance for initial testing or as an add-on test following non-TB diagnosis based on negative sputum smears [24,25]. The results of this study indicated that, during the study period, GeneXpert MTB/RIF was used for 1443 cases among the smear-negative PTB-suspected cases as an "add-on test" to increase the physicians' assurance in making the diagnosis of TB. Among the different CXR subgroups, 449 patients had positive results for MTB based on GeneXpert MTB/RIF. This led the physicians to make the diagnosis of PTB without hesitation, and all but one case were treated accordingly. In such circumstances, it could be said that the physicians used GeneXpert MTB/RIF as a "rule-in test" [26]. In the past, there were several instances in which TB was overdiagnosed or unnecessary TB treatment was provided, which were frequently observed in private clinics/hospitals [27,28]. A bold policy on GeneXpert MTB/RIF accessibility is crucial to reduce the number of overdiagnosed cases that result in unnecessary treatment. A study in San Francisco, USA, claimed that GeneXpert MTB/RIF could greatly reduce the frequency and impact of unnecessary empiric treatment or overtreatment by 94% as well as reduce the impact of transmission in household contact, providing substantial patient and programmatic benefits if used for making management decisions [27]. A cross-sectional study on the use of Xpert-MTB/RIF assay for diagnosis in Southern Ethiopia reported a prevalence of 26.8% among 384 suspected MTB patients [29].

In contrast, among the cases in which MTB was not detected via GeneXpert MTB/RIF in the four CXR subgroups, some were treated as cases of PTB. These reflected the diagnosis of the physicians that were based on other evidence, such as CXR and clinical symptoms, rather than GeneXpert MTB/RIF. In such circumstances, it could be said that the physicians did not completely rely on GeneXpert MTB/RIF as a "rule-out test." A study in the Philippines showed a higher rate of registered patients for PTB treatment among patients with presumptive TB with negative smears and negative GeneXpert MTB/RIF results (89/152 cases, 58.5%). The physicians in the Philippine also diagnosed and treated PTB cases based on clinical diagnosis and other evidence regardless of the negative GeneXpert MTB/RIF results [30]. As claimed in Part II of this study, a physician said *"I usually consider clinical signs together with the results of GeneXpert to start TB treatment,"* while another mentioned that *"GeneXpert is a facilitating tool to confirm TB diagnosis among CXR abnormal cases."* These results confirmed the notation of the Centers for Disease Control and Prevention (CDC) that GeneXpert MTB/RIF assay should be interpreted along with clinical, radiographic, and other laboratory findings; therefore, it cannot replace the need for phenotypic conventional methods [31,32].

Since its endorsement by the WHO, GeneXpert MTB/RIF has been recommended for detecting RR [23,33]. In areas with a high prevalence of RR, GeneXpert MTB/RIF-based detection could be a representation of MDR-TB [24,34–36]. According to the recommendations of CDC, RR is a predictor of MDR-TB because RR coexists with isoniazid (INH) resistance, and thus rapid RR diagnosis would enable the patients with TB to begin successful MDR-TB

treatment much earlier than they would if they were awaiting results from other drug susceptibility tests [30]. In this study, we collected data from patients at high risk of MDR-TB who had CXR results indicative of PTB. GeneXpert MTB/RIF was used for screening 587 cases with CXR results suggesting high risk for MDR-TB, 408 cases with positive sputum smears, and 179 with negative results. Among the smear-positive cases, MTB was detected via GeneXpert MTB/RIF in approximately 13% cases, and further testing via either LPA or conventional TB culture/DST was performed in all cases. Only one case in this group showed discordance between the two tests, i.e., conventional DST reported RR negative, but GeneXpert MTB/RIF reported RR positive. Among 179 patients who were initially smear negative, about 7% were RR positive; subsequent tests also showed high concordance with the previous test. These results corroborate those of previous studies [33,36], in which GeneXpert MTB/RIF could help the physician in the early diagnosis of RR and thereby early and prompt initiation of MDR-TB treatment for patients with positive smears and MDR-TB risk. For patients with smear-negative sputum, GeneXpert MTB/RIF could also detect RR positive cases, which may go undetected in direct smear test alone. As suggested in the literature, the early initiation of an MDR-TB treatment regimen is beneficial and, while the traditional test, using either the liquid or solid medium technique, requires more time to report the DST result to the physician [33,35,36], GeneXpert MTB/RIF allows treatment to be initiated without delay.

The increase in treatment initiation is similar to what was observed in a nationwide retrospective cohort study in Johannesburg, South Africa, which was reported in 2011 and 2013, when GeneXpert MTB/RIF was implemented across districts and provinces. The results showed that the proportion of patients in whom appropriate TB resistance treatment was initiated increased from 6% to 19% [37]. Some studies found that the usage of GeneXpert MTB/RIF among patients at risk of drug-resistant TB was low [38–41]. The results of this study, however, were the opposite as GeneXpert MTB/RIF established at the point of care was used for all high-risk patient groups. The qualitative responses among the healthcare workers in this study indicated their satisfaction with the use of GeneXpert MTB/RIF as it is helpful for MDR-TB prediction. Most physicians preferred to use GeneXpert MTB/RIF for the diagnosis of RR. However, some physicians claimed that they would not use GeneXpert MTB/RIF without confirmation by conventional laboratory tests.

Hence, the decision to start initial MDR-TB treatment was made early, because of which the majority of resistant TB diagnosis were established and initial MDR-TB treatment had been started after RR was detected via GeneXpert MTB/RIF. However, few physicians said that they were considering a false-positive result of RR from GeneXpert MTB/RIF. This needs to be confirmed by LPA or TB culture for MDR-TB treatment.

GeneXpert MTB/RIF had sensitivity of 76% and specificity of 98% in 14 experiments with 5719 samples as an add-on examination following negative smear microscopy results [36]. In this study, the sensitivity of GeneXpert MTB/RIF was also high 85.1% among smear-negative and 100% among MDR-TB diagnosis. In a study in Nepal, a total of 173 sputum samples were collected and processed by microscopy followed by GeneXpert MTB/RIF assay and culture. That study reported the sensitivity, specificity, positive predictive value, and negative predictive values of GeneXpert MTB/RIF assay for smear-negative sputum samples were 74.3%, 96.6%, 86.7%, and 92%, respectively [42]. In contrast, the specificity was rather low 66.0% for PTB and 52.8% for MDR-TB. The discordance between the results of GeneXpert MTB/RIF and conventional tests was 21.8% for smear-negative cases and 41.1% for MDR-TB diagnosis. Likewise, negative smears with positive GeneXpert MTB/RIF results were reported in a study conducted in a hospital in India (34.9%) and in a study conducted in two hospitals in the Philippines (31.5%). The incidence of overdiagnosis will be greatly reduced when a decision is made based on the results of GeneXpert MTB/RIF [30,43]. As noted in the in-depth interview,

*"It makes things easier to consider TB disease even when the patients have no signs or symptoms"* and *"the GeneXpert could help as a tool to reveal the TB strains even when the direct smear is negative, so we do not miss a case."*

The use of GeneXpert MTB/RIF resulted in reducing the TAT and TrT in both smear-negative and high-risk patients. While GeneXpert MTB/RIF, which has high sensitivity, took a day for PTB detection and MDR-TB diagnosis, traditional culture took approximately 1080–1296 hours (45–54 days) and an additional 1204–1464 hours (50–61 days) for conventional DST (if needed). Similar statistics was observed in several other countries, in which the results from GeneXpert MTB/RIF were available within 2 hours (1 day), whereas other conventional techniques took several weeks due to time-consuming and laborious procedures [42–46]. In addition, GeneXpert MTB/RIF could provide information on RR in a remarkably shorter time [44,45,47]. For time to MDR-TB treatment initiation, GeneXpert MTB/RIF took about 4 hours (IQR 3–6 hours) for direct smear-negative cases and 6 hours (IQR 5–7 hours) for high-risk MDR-TB cases. Again, similar to the results from other countries, such as South Korea, Latvia, and South Africa [48,49,50], conventional methods took much longer times than GeneXpert MTB/RIF. The qualitative analysis supported the use of GeneXpert MTB/RIF as a novel test to detect RR considering that the healthcare workers expressed their satisfaction with GeneXpert MTB/RIF testing, which had improved the detection rate and time to appropriate treatment for drug-resistant cases.

Despite the fact that GeneXpert MTB/RIF had proved beneficial for the detection and treatment of patients with PTB and MDR-TB, the limitation of its use at the point of care remains. In certain point of care facilities, there were only GeneXpert MTB/RIF machines with 4 modules, which are capable of only 12 tests per day, thereby affecting the quality of specimens due to backlogged workload. Considering the number of specimens sent from general and community hospitals to central hospitals (study sites), such burdens led to the inability to perform testing efficiently and on a timely basis. Most physicians and nurses expressed their need for the GeneXpert MTB/RIF machine with 16 modules at the point of care.

## Strengths and limitations

This study employed mixed methods and both quantitative and qualitative information to support the results of the study. The study reviewed medical records from different general and central hospitals with large sample sizes, such that all eligible subjects with TB and MDR-TB diagnosis were included. One limitation in the medical chart review was the incomplete information in the medical records, which was either due to human error or because some tests, such as smear microscopy or culture and DST, were not performed. This resulted in inconsistent sample sizes for the results of certain tests. In addition, the retrospective nature of the study using routinely collected data has effect on the data quality. Some key information was missing, such as the timeline of TAT and TrT. Despite these limitations, this study demonstrated the need for continued operational research on the effects of GeneXpert MTB/RIF and other factors associated with TB and MDR-TB treatment outcomes, which might have an impact on TB control programs at large.

## Conclusion and recommendations

GeneXpert MTB/RIF was used as an "add-on test" or a "follow-on test" to aid physicians' decision-making in the diagnosis of PTB and the initiation of MDR-TB treatment. The physicians used it as a "rule-in test" for all patients with negative results of CXR and smears but positive results of GeneXpert MTB/RIF for MTB, leading to timely treatment. On the other hand, about one-fourth of patients with negative results of CXR, smears, and GeneXpert MTB/RIF

were diagnosed with PTB by the physicians, who did not completely rely on GeneXpert MTB/ RIF as a "rule-out test," but rather based their decisions on other evidence, such as CXR and clinical symptoms. The use of GeneXpert MTB/RIF was beneficial for physicians owing to the early detection of RR, leading to prompt treatment of approximately 12% of patients with high risk of MDR-TB (initial positive sputum smear cases and subsequent MTB-RR detected) and about 7% of patients with high risk of MDR-TB who might have otherwise not received proper treatment (initial negative sputum smear cases and subsequent MTB-RR detected). Compared with conventional methods, GeneXpert MTB/RIF effectively reduced the time required for the initiation of second-line drug treatment. Despite its high sensitivity for detecting PTB and MDR-TB, when compared with conventional methods, GeneXpert MTB/RIF demonstrated contradicting results (false positive and/or false negative) in 21.8% of smear-negative patients and 41.1% of patients with high risk of MDR-TB. Thus, the physicians still used the results of other conventional tests (such as culture, LPA, and DST) in their decision-making process.

## Supporting information

**S1 File. Questionnaires (English) Part I.**
(PDF)

## Acknowledgments

The authors would like to thank the directors and data controllers at all of the hospitals in the study areas for their cooperation and assistance in extracting the data needed for analysis in this study. We would like to thank physicians and nurses who provided valuable information during the in-depth interviews.

## Author Contributions

**Conceptualization:** Nathakorn Pongpeeradech, Yuthichai Kasetchareo, Charoen Chuchotta-worn, Saranath Lawpoolsri, Udomsak Silachamroon, Jaranit Kaewkungwal.

**Data curation:** Nathakorn Pongpeeradech, Jaranit Kaewkungwal.

**Formal analysis:** Nathakorn Pongpeeradech.

**Investigation:** Nathakorn Pongpeeradech.

**Methodology:** Nathakorn Pongpeeradech.

**Resources:** Nathakorn Pongpeeradech.

**Software:** Nathakorn Pongpeeradech.

**Supervision:** Jaranit Kaewkungwal.

**Visualization:** Yuthichai Kasetchareo, Jaranit Kaewkungwal.

**Writing – original draft:** Nathakorn Pongpeeradech.

**Writing – review & editing:** Saranath Lawpoolsri, Udomsak Silachamroon, Jaranit Kaewkungwal.

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
