## [Decision Letter · Decision Letter 0]

16 May 2022

PONE-D-22-01562Evaluation of the use of GeneXpert MTB/RIF in a zone with high burden of tuberculosis in ThailandPLOS ONE

Dear Dr. Pongpeeradech,

Thank you for submitting your manuscript to PLOS ONE. After careful consideration, we feel that it has merit but does not fully meet PLOS ONE’s publication criteria as it currently stands. Therefore, we invite you to submit a revised version of the manuscript that addresses the points raised during the review process.

We look forward to receiving your revised manuscript.

Kind regards,

Dwij Raj Bhatta, PhD

Academic Editor

PLOS ONE

Journal Requirements:

4. We note that Figure S1 in your submission contain [map/satellite] images which may be copyrighted. All PLOS content is published under the Creative Commons Attribution License (CC BY 4.0), which means that the manuscript, images, and Supporting Information files will be freely available online, and any third party is permitted to access, download, copy, distribute, and use these materials in any way, even commercially, with proper attribution. For these reasons, we cannot publish previously copyrighted maps or satellite images created using proprietary data, such as Google software (Google Maps, Street View, and Earth). For more information, see our copyright guidelines: http://journals.plos.org/plosone/s/licenses-and-copyright.

a. You may seek permission from the original copyright holder of Figure S1 to publish the content specifically under the CC BY 4.0 license.  

Additional Editor Comments:

Manuscript has been reviewed thoroughly and requires minor revision !Please adress the comments and queries from reviewers

Reviewers' comments:

Reviewer's Responses to Questions

**Comments to the Author**

1. Is the manuscript technically sound, and do the data support the conclusions?

Reviewer #1: Yes

Reviewer #2: Yes

2. Has the statistical analysis been performed appropriately and rigorously? 

Reviewer #1: Yes

Reviewer #2: I Don't Know

3. Have the authors made all data underlying the findings in their manuscript fully available?

Reviewer #1: Yes

Reviewer #2: Yes

4. Is the manuscript presented in an intelligible fashion and written in standard English?

Reviewer #1: Yes

Reviewer #2: Yes

5. Review Comments to the Author

Reviewer #1: 1.Please provide details of data collection procedures including collection of secondary data, interview of physicians and nurses.

2.In ethical considerations, also mention written consent procedure from the participants before data collection.

3.How did you define suspected TB cases? Among suspected cases, 21.8% had no signs and symptoms related to TB. How were they suspected of TB since they had no signs and symptoms?

4.Please summarized the opinions of clinicians on advantages, disadvantages and limitations of GeneXpert MTB/RIF to few sentences, and keep only one or two verbatim.

Reviewer #2: The manuscript is well written. Results are presented in little confusing manner and are need to re-framed in more appropriate pattern. References can be updated with more recent research studies. Overall manuscript is satisfactory.

6. PLOS authors have the option to publish the peer review history of their article (what does this mean?). If published, this will include your full peer review and any attached files.

Reviewer #1: **Yes: **Megha Raj Banjara

Reviewer #2: **Yes: **Dharm raj Bhatta

---

## [Author Response · Author response to Decision Letter 0]

14 Jun 2022

Dwij Raj Bhatta, PhD

Academic Editor

PLOS ONE

---

## [Editor Report · Decision Letter 1]

24 Jun 2022

Evaluation of the use of GeneXpert MTB/RIF in a zone with high burden of tuberculosis in Thailand

PONE-D-22-01562R1

Dear Dr. Kaewkungwal,

We’re pleased to inform you that your manuscript has been judged scientifically suitable for publication and will be formally accepted for publication once it meets all outstanding technical requirements.

Kind regards,

Dwij Raj Bhatta, PhD

Academic Editor

PLOS ONE
---

## [Editor Report · Acceptance letter]

4 Jul 2022

PONE-D-22-01562R1 

Evaluation of the use of GeneXpert MTB/RIF in a zone with high burden of tuberculosis in Thailand 

Dear Dr. Kaewkungwal:

I'm pleased to inform you that your manuscript has been deemed suitable for publication in PLOS ONE. Congratulations! Your manuscript is now with our production department. 

Kind regards, 

on behalf of

Professor Dwij Raj Bhatta 

Academic Editor

PLOS ONE